# Protocol for a cluster randomised controlled trial of the DAFNE*plus* (Dose Adjustment For Normal Eating) intervention compared with 5x1 DAFNE: a lifelong approach to promote effective self-management in adults with type 1 diabetes

Elizabeth Coates [ID],[1] Stephanie Amiel [ID],[2] Wendy Baird,[1] Mohammed Benaissa,[3] Alan Brennan,[1] Michael Joseph Campbell [ID],[1] Paul Chadwick,[4] Tim Chater,[1] Pratik Choudhary,[2] Debbie Cooke,[2] Cindy Cooper,[1] Elizabeth Cross,[1] Nicole De Zoysa,[5] Mohammad Eissa,[3] Jackie Elliott,[6] Carla Gianfrancesco,[7] Tim Good,[3] David Hopkins,[8] Zheng Hui,[3] Julia Lawton,[9] Fabiana Lorencatto,[4] Susan Michie,[4] Daniel John Pollard [ID],[1] David Rankin,[9] Jose Schutter,[10] Elaine Scott,[1] Jane Speight,[11] Stephanie Stanton-Fay,[4] Carolin Taylor,[7] Gillian Thompson,[12] Nikki Totton,[1] Lucy Yardley,[13] Aleksandr Zaitcev,[3] Simon Heller,[6] on behalf of The DAFNE*plus* group

► Prepublication history and additional materials for this paper is available online. To view these files, please visit the journal online (http://dx.doi.org/10.1136/bmjopen-2020-040438).

For numbered affiliations see end of article.

**Correspondence to**
Professor Simon Heller;
s.heller@sheffield.ac.uk

## ABSTRACT

**Introduction** The successful treatment of type 1 diabetes (T1D) requires those affected to employ insulin therapy to maintain their blood glucose levels as close to normal to avoid complications in the long-term. The Dose Adjustment For Normal Eating (DAFNE) intervention is a group education course designed to help adults with T1D develop and sustain the complex self-management skills needed to adjust insulin in everyday life. It leads to improved glucose levels in the short term (manifest by falls in glycated haemoglobin, HbA1c), reduced rates of hypoglycaemia and sustained improvements in quality of life but overall glucose levels remain well above national targets. The DAFNE*plus* intervention is a development of DAFNE designed to incorporate behavioural change techniques, technology and longer-term structured support from healthcare professionals (HCPs).

**Methods and analysis** A pragmatic cluster randomised controlled trial in adults with T1D, delivered in diabetes centres in National Health Service secondary care hospitals in the UK. Centres will be randomised on a 1:1 basis to standard DAFNE or DAFNE*plus*. Primary clinical outcome is the change in HbA1c and the primary endpoint is HbA1c at 12 months, in those entering the trial with HbA1c >7.5% (58 mmol/mol), and HbA1c at 6 months is the secondary endpoint. Sample size is 662 participants (approximately 47 per centre); 92% power to detect a 0.5% difference in the primary outcome of HbA1c between treatment groups. The trial also measures rates of hypoglycaemia, psychological outcomes, an economic evaluation and process evaluation.

### Strengths and limitations of this study

► Comparison of group therapy against another group therapy will standardise the treatment comparison.
► Cluster randomisation to avoid contamination of the intervention material.
► Number of sites in both England and Scotland representing a wide range of National Health Service Trusts.
► Use of a covariate constrained methodology to randomise means that sites are matched which can create issues if sites drop out.
► Blinding not possible in trial due to the intervention and design.

**Ethics and dissemination** Ethics approval was granted by South West-Exeter Research Ethics Committee (REC ref: 18/SW/0100) on 14 May 2018. The results of the trial will be published in a National Institute for Health Research monograph and relevant high-impact journals.
**Trial registration number** ISRCTN42908016.

## INTRODUCTION
### Background and rationale
Type 1 diabetes (T1D) is characterised by absolute insulin deficiency, requiring insulin to be injected subcutaneously several times a day. Successful management requires those affected (>300 000 adults in the UK)[1] to keep their blood glucose levels sufficiently close

to recommended targets to avoid long-term complications including blindness, renal failure, amputations and premature death.[2] In addition, exogenous insulin therapy can prevent high blood glucose and acute, life-threatening emergencies such as diabetic ketoacidosis, as well as being a tool to prevent long-term complications.

Achieving the blood glucose control to help prevent complications depends on an individual's ability to self-manage their condition, calculating precise insulin doses based on accurate estimations of food intake before every meal, frequent blood glucose measurements and account for fluctuations in physical activity, illness and hormones. If people with T1D are unable or unwilling to calculate and administer their insulin doses correctly, their blood glucose either runs high, increasing the risks of complications or else falls too low leading to hypoglycaemia. Hypoglycaemia, if severe, can result in acute cognitive impairment, confusion, collapse and injury, coma or even death.[3] Thus, people with T1D must acquire complex self-management knowledge and skills, and have the motivation and ability to apply them effectively every day. The responsibility of diabetes healthcare professionals (HCPs) is to ensure that all people with T1D have the opportunity to acquire these skills and are supported in applying them successfully in everyday life.

'Dose Adjustment For Normal Eating' (DAFNE) is a structured education programme run within the National Health Service (NHS), designed to enable adults with T1D to learn or enhance their self-management skills in flexible intensive insulin therapy to improve both glucose control and quality of life. It is a five-day training course, delivered in small groups. DAFNE has been delivered to over 51 000 adults in the UK.[4] The publication of the UK DAFNE randomised controlled trial (RCT) in 2002[5] established the ability of structured education courses to enable people with diabetes to acquire the knowledge and skills to live successfully with this lifelong condition. The subsequent rollout of DAFNE across the UK has enabled many individuals to meet these demands and achieve their goals, but over half of DAFNE graduates still struggle to manage glucose levels consistently. After attending a DAFNE course, people have better quality of life, better control of blood glucose levels and are admitted to hospital less often for diabetes emergencies.[6] Many DAFNE graduates find the course helpful; quality of life improves and rates of severe hypoglycaemia fall. However although glycated haemoglobin (HbA1c) falls and in one trial, this improvement was sustained for 2 years, average HbA1c, the intermediate measure of glucose control that best predicts risk of diabetes complications, remains well above recommended UK targets.[7 8] Many find it difficult to implement and sustain the skills needed to maintain blood glucose levels and often struggle to obtain suitable support from HCPs.[6 9–15]

The DAFNE*plus* intervention has been developed through modifying the existing DAFNE programme by incorporating techniques for initiating and sustaining behaviour change, and supplementing this with structured follow-up support and enhanced information technology. The aim of this trial is to investigate whether the DAFNE*plus* programme will produce improved and sustainable diabetes self-management behaviour and better glucose outcomes than currently achieved with standard DAFNE, without compromising quality of life in the longer term.

## Aims and objectives

The primary aim of this study is to conduct a cluster RCT comparing the new DAFNE*plus* intervention to the existing DAFNE programme to answer the following question:

In adults with T1D, will modifying the existing DAFNE programme and developing structured professional input, using learning from our recent research, behavioural change theory and new forms of technological support, produce improved and sustained diabetes self-management behaviours, leading to better glucose control than currently achieved, using the existing DAFNE intervention, without compromising quality of life?

The primary objective is to assess the effects of the intervention on glycaemic control, as measured by HbA1c at 12 months.

The secondary objectives of this trial are:

1. To compare the effects of the intervention (DAFNE*plus*) to standard DAFNE on diabetes-specific quality of life.
2. To compare the medium term effect of the intervention (DAFNE*plus*) to standard DANFE on glycaemic control as measured by HbA1c using data at 6 months.
3. To compare the effects of the intervention (DAFNE-*plus*) to standard DAFNE on other biomedical outcomes.
4. To compare the effects of the intervention (DAFNE-*plus*) to standard DAFNE on psychological outcomes.
5. To undertake a mixed methods process evaluation to aid understanding of the RCT findings, and to inform decision making about the implementation of DAFNE-*plus* in clinical care post-trial.
6. To assess fidelity of delivery of the DAFNE*plus* intervention.
7. To undertake a health economic analysis to determine the cost-effectiveness of DAFNE*plus* versus standard DAFNE.

## METHODS AND ANALYSIS
### Trial design

The study will use a pragmatic cluster RCT design. This is required since 'contamination' of the control arm may occur if DAFNE HCPs, trained in the new programme were to deliver standard DAFNE. Hence the randomisation of DAFNE centres rather than individuals.[16] Figure 1 shows the flow of participants through the trial (see online supplemental material 1 for WHO Trial Registration Data Set).

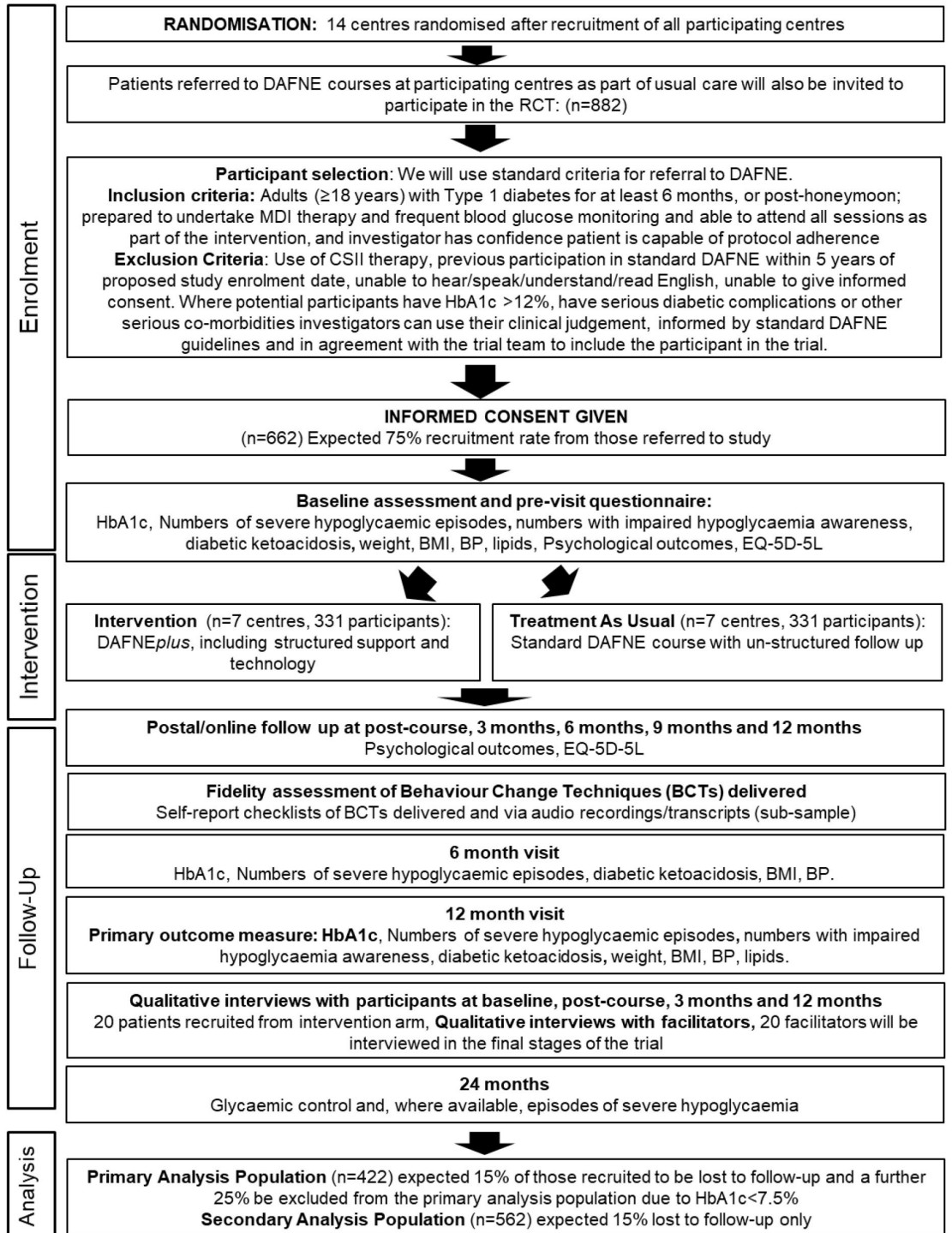

**Figure 1** RCT flow diagram. BMI, body mass index; BP, blood pressure; CSII, continuous subcutaneous insulin infusion; DAFNE, dose adjustment for normal eating; EQ-5D, EuroQoL-5 Dimension 5 Level; HbA1C, glycated haemoglobin; MDI, multiple daily injection; RCT, randomised controlled trial.

## Study setting

The trial will be delivered in adult diabetes centres in secondary care NHS hospitals in the UK. The eligibility criteria for study centres are:

1. Diabetes centre delivering DAFNE to adults with T1D.
2. At least three DAFNE educators trained in delivering the 5-week model of DAFNE.
3. Delivery of sufficient DAFNE courses per year to recruit study sample.

Adults with T1D eligible for or referred to DAFNE courses at participating centres as part of usual care will be eligible to be invited to participate in the RCT, and standard criteria for referral to DAFNE will be used.

## Eligibility criteria
### Inclusion criteria
1. Adults (≥18 years).
2. Diagnosis of T1D for at least 6 months, or posthoneymoon. The honeymoon period refers to the time when, postdiagnosis, people start taking insulin injections, and their insulin producing cells sometimes recover temporarily (generally around 3–12 months. The dose of insulin needed might reduce during this period, and some people might even need to stop using insulin for a while, but eventually it will be needed again. The criteria for referral to DAFNE at least 6 months after diagnosis is to allow for the honeymoon period to have passed before attendance at the course.
3. Prepared to undertake multiple daily injection therapy.
4. Prepared to undertake frequent self-monitoring of blood glucose.
5. Confirms availability to attend all sessions as part of the intervention.
6. Investigator has confidence that the patient is capable of adhering to all the trial protocol requirements.

### Exclusion criteria
1. Current use of continuous subcutaneous insulin infusion pump therapy.
2. HbA1c >12% (108 mmol/mol) (Investigators can use their judgement, informed by standard DAFNE guidelines and in agreement with the trial team, to include participants with HbA1c >12%).
3. Serious diabetic complications (eg, blindness, renal dialysis). (Investigators can use their clinical judgement, informed by standard DAFNE guidelines and in agreement with the trial team).
4. Other serious comorbidities, for example, psychosis, diagnosed eating disorder. (Investigators can use their clinical judgement, informed by standard DAFNE guidelines and in agreement with the trial team).
5. Previous participation in standard DAFNE course less than five years before proposed study enrolment date.
6. Unable to speak/hear/understand/read or write in English.
7. Unable to give written informed consent.

## Recruitment
Patient participants will be identified from current caseloads of adults with T1D from each participating centre. They will be sent an invitation letter and information sheet before the course. A member of the clinical team in participating centres will then telephone potential participants to discuss whether or not they are interested in principle in taking part. If interested, they will be asked to consent to participate at their baseline visit. In both trial arms, if they do not want to take part in the research they will be offered attendance at a standard DAFNE course that is not part of this trial, if that is their wish. Reasons for non-participation in the trial will be recorded.

In order to maximise recruitment to the courses, a reserve list of eligible patients will be held at participating centres. Eligible patients may also be invited to take part by their HCP during routine face-to-face appointments or via telephone. Trial information meetings may also be held during the recruitment period at various locations in centres.

Written informed consent will be obtained from all participants. Members of the local study teams will be responsible for taking informed consent from potentially eligible study participants at the DAFNE centres. The process for obtaining participant informed consent will be in accordance with the REC guidance, and Good Clinical Practice (GCP) and any other regulatory requirements that might be introduced.

Written informed consent to contribute to the process evaluation will also be taken from HCPs in participating sites by the central study team.

## Interventions
### Standard DAFNE (control arm)
DAFNE is a skill-based structured education programme for adults with T1D delivered in the NHS. Two evidence-based models of delivering standard DAFNE are in operation, whereby the five sessions of the course are delivered weekly or daily, as described elsewhere.[17] Each course is delivered to seven participants on average (minimum of four and maximum of eight). Standard DAFNE will be delivered, as usual care, by trained DAFNE educators in the NHS, including diabetes specialist nurses, dietitians and physicians.

The aim of the course is to train adults with T1D in the skills to manage their condition effectively. It covers numerous topics in a progressive modular based structure. In addition to the five days of the course, participants are asked to attend a baseline appointment before the DAFNE course, and they are also typically invited to attend an optional group follow-up session 6–8 weeks after the course. They may also attend routine appointments every 6–12 months and seek ad hoc support from local diabetes clinicians post-course.

For the purposes of this study, the control arm will be the 5-week model of standard DAFNE to match the frequency of sessions offered in DAFNE*plus*. All participants in the control arm will be given access to a standalone bolus calculator to assist them with calculating insulin doses. There will be no structured follow-up appointments beyond those provided in usual care. To qualify as adherent for statistical purposes, participants need to have attended the equivalent of 4 days of the course including days one and two which are mandatory; it will be acceptable to include half days in the total.

### DAFNE*plus* (intervention arm)
DAFNE*plus* will be delivered by trained DAFNE educators in the NHS. In DAFNE*plus*, those delivering the intervention are referred to as 'facilitators', as opposed to 'educators' in standard DAFNE. These will be HCPs including

diabetes specialist nurses, dietitians and physicians, all of whom will be using DAFNE principles as an integral part of the management of T1D in adults. DAFNE*plus* is a complex intervention, defined by the Medical Research Council[18] as having 'several interacting components', described in summary below.

The development of the content and structure of the DAFNE*plus* programme was informed by the Behaviour Change Wheel framework.[19] The intervention's proposed functions are served by behaviour change techniques (BCTs), specified in the hierarchical BCT Taxonomy V.1,[20] deemed its 'active ingredients'.[21] The development of the DAFNE*plus* programme (described in [22]) was informed by expert consensus, integrating data on participant-generated and clinician-generated barriers and facilitators to sustaining DAFNE with the findings from a synthesis of qualitative evidence about post-DAFNE challenges.[22] Prior to this RCT, the DAFNE*plus* programme was piloted in three NHS Diabetes Centres.

The DAFNE*plus* programme comprises three components:

### DAFNE*plus* course
The group-based course component of the DAFNE-*plus* programme is delivered one day per week, over five consecutive weeks, and is based on a revision of the standard DAFNE 5-week curriculum, with a view to strengthening and sustaining self-management behaviours over a longer term to enable them to achieve blood glucose levels closer to target. Participants will attend an individual precourse appointment approximately two weeks before the course which serves as their introduction to the programme, during which they are given access to and trained in using the DAFNE*plus* technology (see below), as well as a bolus calculator.

New sessions included in the DAFNE*plus* course include technology assisted individual review, emotional aspects of living with diabetes and its management, harnessing social support and behavioural change—including additional support for action planning and relapse prevention to help participants achieve their self-management goals. The curriculum was revised to be consistent with modern approaches to the recommended language used in diabetes care.[23] Requirements to qualify as adherent for statistical purposes are defined above.

### Structured follow-up support
The model of structured follow-up support builds on the clinical and behavioural skills introduced during the course to enable participants to maximise the efficacy of key DAFNE*plus* principles to improve self-management and achieve/sustain glycaemic targets. As part of the trial, up to five one-to-one consultations (face-to-face, telephone or in some centres, web-based video calling) with a DAFNE facilitator will be offered, delivered at progressively wider spaced intervals during the 12 months after the course. Appointments are supported by paperwork to 'activate' both the participant and the facilitator prior to meeting.

The purpose of these individual sessions is to review participants' progress with managing their diabetes, including progress with their action plans, review blood glucose data on the DAFNE*plus* website, revise course material, address any additional clinical needs and sign-post participants to any relevant sources of support. In addition, ad hoc support by telephone, email or web-based video calling will be available, as necessary. To qualify as adherent for statistical purposes, participants will need to have attended a minimum of three follow-up sessions.

### Digital technology
The DAFNE*plus* programme incorporates two forms of digital technology via the DAFNE*plus* website and box. Participants will be given access and training at the pre-course appointment, so that they can use the technology before and throughout the 12-month programme. The DAFNE*plus* box (Withcare+) transmits, stores and displays blood glucose (and other) data on a secure-server via the DAFNE*plus* website in formats to help people with T1D and their HCPs recognise and interpret blood glucose patterns. The website also includes an e-learning section to help maintain knowledge of the DAFNE*plus* approach.

### Training and supervision
A clinical psychologist who specialises in diabetes and is experienced in training diabetes professionals in behavioural change skills will lead the development and delivery of DAFNE*plus* facilitator training and supervision. The training programme is delivered over a maximum of five days and will build on the existing skill-set of DAFNE facilitators but also draw on additional behavioural science to deliver the revised curriculum.

Throughout the trial, facilitators in each centre will be offered supervision by the clinical psychologist and a DAFNE*plus* facilitator. Supervision will comprise weekly teleconferences before and during the first DAFNE-*plus* course, weekly email supervision (for subsequent courses) and ad hoc remote support to allow issues that arise to be addressed in a timely manner during the trial.

### Criteria for withdrawal from or discontinuation of trial treatment
The decision regarding participation in the study is entirely voluntary, and consent regarding study participation may be withdrawn at any time without affecting the quality or quantity of future medical care. No study-specific interventions will be undertaken before informed consent has been obtained.

A participant will be classed as complete if they have continued in the study until the last protocol defined intervention (final 12-month outcome assessment), although there may be missing data for individual participants.

## Random allocation

On recruitment of centres and following ethical approval, the participating centres will be randomised on a 1:1 basis to control or the intervention arm of the trial by the trial statistician. As there are numerous stratification variables that have been identified as clinically important and the small number of randomising centres, a covariate constrained methodology[24] will be employed. The centres will be matched on the number of patients within the centre, number of educators within the centre and number of previous DAFNE courses delivered (as a marker of centre experience) to balance centres between the two arms of the trial.

## Blinding

Due to the nature of the intervention, it is not possible for members of the study team working directly with participants or the intervention to be blinded. Additionally, the blinding of the statistician is problematic due to the cluster level randomisation. Statisticians are usually involved within Trial Management Group (TMG) discussions and have access to status reports where the potential for unintentional unblinding is a high possibility. It is considered important for the statistician to be included in these aspects of the trial management and so after discussion with senior statisticians at the Clinical Trials Research Unit (CTRU) and the independent statistician on the Trial Steering Committee (TSC), it has been deemed acceptable that the statisticians are not blind within this study.

## Outcomes

Table 1 shows a breakdown of all outcome measures.

## Biomedical outcomes

The primary biomedical outcome is an integrated measure of glucose levels over the previous 4–6 weeks, defined by HbA1c (using a centralised assay to ensure standardisation). The primary endpoint is HbA1c at 12 months, in those entering the trial with HbA1c >7.5% (58 mmol/mol), and HbA1c at 6 months is the secondary endpoint.

Our primary aim is to compare HbA1c between the two arms and we have therefore confined our primary analysis to those with raised A1c values at baseline. We, therefore, excluded those with an HbA1c below 7.5% (58 mmol/mol) when calculating the primary endpoint as these people have less need to reduce their HbA1c.

However, we have included participants with lower A1c values to ensure we can calculate important secondary outcomes part rates of hypoglycaemia, and other biomedical and psychological outcomes. We have estimated the expected proportion of participants with A1c values above 7.5% at 75% of those currently undertaking DAFNE courses based on a national research database.

Other secondary outcomes are the number of participants achieving either an HbA1c <7.5% (58 mmol/mol) or a decrease in HbA1c of ≥0.5% (≥5.5 mmol/mol)

which will be calculated at both 6 and 12 months post course. These cut-off points are recognised throughout the diabetes research community as being clinically relevant.[25] We will also collect and analyse 24-month outcome data (HbA1c and severe hypoglycaemic episodes) and analyse after the main study has closed and been reported based on locally available clinical data which is routinely collected annually in clinical centres.

Other secondary biomedical outcomes will include: Severe hypoglycaemia, as defined by the American Diabetes Association,[26] denotes severe cognitive impairment requiring external assistance for recovery, both rates and proportion of those affected; Diabetic ketoacidosis, both rates and proportion of those affected; weight; body mass index; blood pressure; lipids; albumin/creatinine ratio.

## Psychological outcomes and process evaluation

### Quantitative outcomes

Psychological outcomes and process measures will be collected via self-completed postal or online questionnaires at baseline, course completion, 3, 6, 9 and 12 months (see table 1).

The primary psychological outcome is the impact of diabetes on quality of life assessed at 12 months using a 15-domain version of the Audit-Dependent Diabetes Quality of Life Questionnaire.[27]

Additional psychological constructs are assessed with validated questionnaires and study-specific individual items, based on: existing knowledge about their association with the trial's primary biomedical outcome (HbA1c) and primary psychological outcome (diabetes-specific quality of life), including the findings of the YourSAY survey[28]; previous work with the DAFNE intervention, and the theoretical framework underpinning the DAFNE*plus* intervention development and possible treatment mechanisms.[19 29 30]

### Qualitative outcomes

Interviews will be undertaken with a subset of participants randomised to the intervention at baseline, course completion, 3 months and 12 months (figure 1) to explore how key elements of the intervention influence and inform changes to, and maintenance of, key self-management behaviours over time. Facilitators will be interviewed from across the intervention sites to explore their experiences of intervention delivery and their views about the training, resourcing and support staff would need to deliver DAFNE*plus* in routine care.

### Fidelity assessment

We will explore fidelity of delivery using two methods to assess the extent to which the intervention content specified in the DAFNE/DAFNE*plus* manuals is delivered as intended: self-report checklists completed by educators/facilitators, and objectively analysed delivery from session audio recordings. Fidelity of delivery will be assessed in standard DAFNE as well as DAFNE*plus* in order to

**Table 1** List of outcome and process measures

| Concepts | Questionnaire | Baseline: precourse appt | Course Completion | Postcourse assessments 3 months* | 6 months* | 9 months* | 12 months* |
|---|---|---|---|---|---|---|---|
| **Demographic/clinical** | | | | | | | |
| Glycaemic control (HbA1c) | N/A | ✓ | | | ✓ | | ✓ |
| Lipids | N/A | ✓ | | | | | ✓ |
| Body mass index (height/weight) | N/A | ✓ | | | ✓ | | ✓ |
| Blood pressure | N/A | ✓ | | | ✓ | | ✓ |
| Episodes of severe hypoglycaemia | N/A | ✓ | | | ✓ | | ✓ |
| Episodes of ketoacidosis | N/A | ✓ | | | ✓ | | ✓ |
| Demographics | Individual items | ✓ | | | ✓ | | ✓ |
| Hypoglycaemia awareness | Gold score[36] and DAFNE hypo awareness measure | ✓ | | | ✓ | | ✓ |
| **Primary psychological outcomes** | | | | | | | |
| Diabetes-specific quality of life | ADDQoL-15[27] | ✓ | | | ✓ | | ✓ |
| **Secondary psychological outcomes** | | | | | | | |
| Diabetes distress | Problem Areas In Diabetes (short-form)[37] | ✓ | | | ✓ | | ✓ |
| Diabetes-specific quality of life | Dawn Impact of Diabetes Profile[38] | ✓ | | | ✓ | | ✓ |
| Diabetes-specific positive well-being | 4-item sub-scale of the Well Being Questionnaire[39] | ✓ | | | ✓ | | ✓ |
| Fear of hypoglycaemia | Hypoglycaemia Fear Survey-11 (short-form)[40] | ✓ | | | ✓ | | ✓ |
| Health status | Health and Self-Management in Diabetes[41] | ✓ | | | ✓ | | ✓ |
| Health status | EQ-5D-5L[42] | ✓ | | | ✓ | | ✓ |
| Healthcare utilisation | Individual items | ✓ | | | ✓ | | ✓ |
| Resource allocation | Individual items | | ✓ | | | | |
| **Process measures** | | | | | | | |
| Diabetes management experiences (satisfaction) | Diabetes Management Experiences Questionnaire[43] | ✓ | ✓ | ✓ | | ✓ | |
| Self-regulatory skills/behavioural regulation | Self-regulation Questionnaire*[44] | ✓ | ✓ | ✓ | | ✓ | |
| Diabetes strengths and resilience | Diabetes Strengths and Resilience Questionnaire[45] | ✓ | ✓ | ✓ | | ✓ | |
| Beliefs about capabilities: diabetes self-care | Confidence in Diabetes Scale*[46] | ✓ | ✓ | ✓ | | ✓ | |
| Beliefs about capabilities: hypoglycaemia confidence | Hypoglycaemia Confidence Scale[47] | ✓ | ✓ | ✓ | | ✓ | |
| Diabetes-specific self-care behaviours | Diabetes Self-Care Behaviours[48] | ✓ | ✓ | ✓ | | ✓ | |
| Beliefs about consequences of engaging in DAFNE behaviours and weaving diabetes management into everyday routines | Individual items* | ✓ | ✓ | ✓ | | ✓ | |
| Evaluation of technology (DAFNEplus website in intervention group and bolus calculator in control group) | System usability scale[49] | | ✓ | ✓ | | ✓ | |

*Description about the development and modifications of these questionnaires and individual items are detailed in online supplemental material 4.
ADDQoL, Audit of Diabetes-Dependent Quality of Life Questionnaire; DAFNE, Dose Adjustment For Normal Eating; EQ-5D-5L, EuroQol-5 Dimension 5 Level; N/A, Not Applicable.

assess any loss of treatment differentiation and potential contamination between the two arms.

1. Self-report checklists: Facilitators will complete checklists after each session. Each checklist lists the components intended to be delivered in each session (according to the manual). These components correspond to different BCTs. Each component will be rated as fully, partially or not delivered, with space for additional comments. The proportion of intended components rated as partially/fully delivered by educators/ facilitators will be calculated, with <50% of intended content delivered classified as low fidelity; 51%–79% as moderate fidelity, and 80%–100%' as high fidelity.[31]
2. Objectively analysed delivery: A subsample of group course sessions in both arms will be audio-recorded and transcribed verbatim. Transcripts will be coded into component BCTs using an established taxonomy.[20] BCTs identified in each session transcript will be compared with corresponding section of the intervention manual that specifies which BCTs are intended to be delivered in that session. Fidelity will be calculated in terms of the percentage of manual-specified BCTs delivered as intended. Additional BCTs delivered that are not specified in the curricula will also be noted.

Detailed plans for the process evaluation are in online supplemental material 2.

## Health economic outcomes

Table 1 details the health economic data collected in the trial. In addition, data collected from the DAFNE*plus* website will be used to cost the intervention. The analysis population for the health economic analyses will include all trial participants, as it is important that the analysis of health economic data includes all participants who would be eligible to receive DAFNE*plus* (if it were to be implemented). In line with the statistical analysis, we will conduct subgroup analyses in participants with an HbA1c ≤7.5% and >7.5% (58 mmol/mol).

Two health economic analyses will be conducted, a primary long-term analysis using the Sheffield T1D Policy Model and a secondary analysis of the data collected in the trial. All health economic analysis will compare the incremental cost-effectiveness ratio of DAFNE*plus* versus DAFNE to standard National Institute for Health and Care Excellence thresholds to determine cost-effectiveness.[32] See online supplemental material 3 for detailed plans for the economic evaluation.

## Safety outcomes

### Adverse events

Study centres are only required to report as adverse events episodes of diabetic ketoacidosis and severe hypoglycaemia which while not requiring admission to hospital have been noted by either the participant or their relative/partner etc. These will be recorded on the data collection form and database.

### Reporting

We do not anticipate many Serious Adverse Events related specifically to DAFNE*plus* or standard DAFNE but will report any which are deemed related to the study intervention and or are unexpected to the Sponsor and the REC in line with best practice.

## Sample size

It is expected that there will be 882 patients referred for DAFNE courses within the 15-month recruitment window and of these it is expected 75% (662 patients) will be recruited, equivalent to 47 participants at each of the 14 centres. From current DAFNE data, a further 25% are expected not to meet the primary analysis population criteria of baseline HbA1c greater than 7.5% (58 mmol/mol), leaving 497 participants. Finally, we anticipate 15% of participants to be lost to follow-up by the 12-month stage, therefore giving a primary analysis population of 422 participants. The sample size takes into account the design effect associated with the cluster design of the study. With an intracluster correlation coefficient (ICC) of 1.5% (from previous DAFNE data) and 30 participants per cluster (422 participants over 14 centres) the design effect is 1.435 leaving the effective total sample size of n=294 participants (n=147 per arm).

Using a two sample comparison of mean HbA1c at the 12-month follow-up with two-sided alpha of 5%, a correlation of 0.5 between baseline and final values and a SD of 1.45 (from previous DAFNE data), the trial sample gives 92% power to detect a 0.5% difference in HbA1c (the minimum clinically important difference) between the two treatment groups in the study.

## Statistical analysis

The primary analysis population will be participants that had an HbA1c greater than 7.5% (58 mmol/mol) at baseline and the analysis will be completed on an intention-to-treat (ITT) basis. This primary analysis is to assess the difference between the two treatment groups on the mean HbA1c at 12 months which will be completed using a multiple linear regression model with coefficients estimated using generalised estimating equations to account for the clustering design. A 95% confidence interval for the difference between the two treatment groups will be presented. Appropriate covariates will be included in the model, along with the participant's baseline HbA1c, to adjust the treatment effect accordingly.

The secondary analysis population is all consenting participants in the trial and analysis will again be completed on an ITT basis. This population will also be used to assess the difference in psychological outcomes between the two treatment groups using the same model as for the primary analysis.

A full statistical analysis plan has been written and was circulated to the TMG and TSC before being signed-off. This is available in online supplemental material 5. All analyses results will be reported according to the revised

Consolidated Standards of Reporting Trials 2010 statement for cluster RCTs.[33]

## Data collection and management

Case report forms will be completed by DAFNE facilitators/educators at each study visit. Follow-up questionnaires will be self-completed by participants at each follow-up point. Participants will be allocated a unique identification number to identify them throughout the trial.

Plans to promote retention and follow-up of all trial participants include research appointments being scheduled and followed up by their clinical teams at 6 months and 12 months. Overdue questionnaires are followed-up with an email reminder and then telephone call from CTRU. All participants received email newsletters to update them on trial progress.

Data will be entered onto the DAFNE*plus* database on CTRU's secure online system, hosted on University of Sheffield servers. Access is restricted such that users can enter and view only information required to perform their role.

Identifiable data will be shared with CTRU and the supporting study team and DAFNE*plus* website teams. Consent will be obtained from the participant for this to occur. Data will be stored securely on access-restricted network drive folders in accordance with CTRU standard operating procedures (SOPs).

All consent forms and questionnaires will be kept in a locked filing cabinet in a secured area and will be retained for a minimum of 5 years after study completion, in accordance with the sponsor's archiving requirements. Sheffield CTRU may request consent forms to be sent from the research site to the CTRU via post or email as part of remote monitoring procedures.

The nature, frequency and intensity of trial monitoring will be outlined in the site monitoring plan, which will be devised in accordance with CTRU SOPs.

## Patient and public involvement

In addition to the patient representation on the trial oversight committees, this trial is supported by a patient advisory group who have and will continue to meet regularly during the conduct of the trial (and the wider programme grant). Patient input has been sought throughout on the trial and intervention design, the informational material to support trial conduct and patient burden.

## Trial oversight committees

Two oversight committees have been established to oversee the conduct of this trial— the TSC and TMG, the composition of each is listed at the end of this paper. A Data Monitoring and Ethics Committee has not been convened, on the grounds that the study is low risk, in line with CTRU SOP GOV003. This has been approved by the Sponsor and TSC.

## ETHICS AND DISSEMINATION

The RCT was not initiated until the protocol, informed consent forms and participant information sheets received approval from the Research Ethics Committee, the Health Research Authority and local Capacity and Capability is confirmed by the respective NHS Research and Development departments. MHRA (Medicines and Healthcare products Regulatory Agency) approval was not required for this study.

The RCT is being conducted in accordance with the ethical principles that have their origin in the Declaration of Helsinki[34]; the principles of GCP, and the UK Framework for Health and Social Care Research.[35]

Outputs from the trial will be generated in accordance with the communication and dissemination strategy. A number of academic outputs will be produced as the data are analysed from the trial. Journals will be selected based on the highest possible impact. Other stakeholder-specific outputs in relevant formats will also be produced for commissioners, third sector and user advocacy organisations.

**Author affiliations**
[1]School of Health and Related Research, The University of Sheffield, Sheffield, UK
[2]Department of Diabetes, King's College London Faculty of Life Sciences and Medicine, London, UK
[3]Department of Electronic and Electrical Engineering, University of Sheffield, Sheffield, UK
[4]Epidemiology and Public Health, UCL, London, UK
[5]King's College Hospital, London, UK
[6]Department of Oncology and Metabolism, The University of Sheffield, Sheffield, UK
[7]Diabetes Centre, Sheffield Teaching Hospitals NHS Foundation Trust, Sheffield, UK
[8]General and Emergency Medicine, King's College London, London, UK
[9]Centre for Population Health Sciences, University of Edinburgh, Edinburgh, UK
[10]ScHARR, University of Sheffield, Sheffield, UK
[11]The Australian Centre for Behavioural Research in Diabetes, Melbourne, Victoria, Australia
[12]Northumbria Healthcare NHS Foundation Trust, North Shields, UK
[13]Academic Unit of Psychology, University of Southampton, Southampton, UK

**Acknowledgements** The authors would like to thank all of the patients who are taking part in this trial, and all the participating clinical sites. The authors would also like to thank Becky Brown for administrative support to the trial, and Chris Turtle for data management.

**Collaborators** Trial Co-ordinating Centre: Elaine Scott, Trial Manager; Liz Cross, Senior Trial Manager; Jose Schutter, Research Assistant; Tim Chater, Data Manager; Becky Brown, Trials Support Officer; Professor Cindy Cooper, Trials Oversight (Sheffield CTRU, University of Sheffield). Trial Management Group: Professor Stephanie Amiel (King's College London); Professor Wendy Baird (University of Sheffield); Dr Mohammed Benaissa (University of Sheffield); Professor Alan Brennan (University of Sheffield); Becky Brown (University of Sheffield); Dr Paul Chadwick (University College London); Tim Chater (University of Sheffield); Dr Pratik Choudhary (King's College London); Dr Debbie Cooke (University of Surrey); Professor Cindy Cooper (University of Sheffield); Dr Mohammad Eissa (University of Sheffield); Dr Tim Good (University of Sheffield); Carla Gianfrancesco (Sheffield Teaching Hospitals NHS Foundation Trust); Dr Jackie Elliott (University of Sheffield); Dr David Hopkins (King's College Hospital NHS Foundation Trust); Professor Julia Lawton (University of Edinburgh); Dr Fabiana Lorencatto (University College London); Professor Susan Michie (University College London); Dan Pollard (University of Sheffield); Dr David Rankin (University of Edinburgh); Jose Schutter (University of Sheffield); Elaine Scott (University of Sheffield); Professor Jane Speight (Deakin University (Australia)); Dr Stephanie Stanton-Fay (University College London); Carolin Taylor (Sheffield Teaching Hospitals NHS Foundation

# Open access

Trust); Gill Thompson (Northumbria Healthcare NHS Foundation Trust); Nikki Totton (University of Sheffield); Chris Turtle (University of Sheffield); Dr Nicole de Zoysa (King's College Hospital NHS Foundation Trust). Trial Steering Committee membership: Chair: Professor Simon Griffin (University of Cambridge); Independent Statistician: Professor Catherine Hewitt (University of York); Independent Clinical Expert: Professor James Shaw (University of Newcastle); Patient Representative: Arthur Durrant, Lynne Dawson; Sponsor's Representative: Dr Erica Wallis (Sheffield Teaching Hospitals NHS Foundation Trust); Funder's Representative: Ramnath Elaswarapu (National Institute for Health Research).

**Contributors** ES, TC, ECr and JSc have responsibility for running the trial and acquisition of the trial data, under the leadership of SH as Chief Investigator. All other authors (ECo, SA, WB, MB, AB, MJC, PC, PCh, DC, CC, NDZ, ME, JE, CG, TG, DH, ZH, JL, FL, SM, DJP, DR, JS, SS-F, CT, GT, NT, LY, AZ and SH) made substantial contributions to the conception or design of the work, whether as coinvestigators and/or as members of the Trial Management Group. ECo prepared the first draft of the manuscript and edited this with input from other authors. All authors provided critical review and approval of the final manuscript.

**Funding** This work is funded by the National Institute for Health Research through their Programme Grants for Applied Research Programme (Ref: RP-PG-0514-20013) and NHS England provide funding for Excess Treatment Costs.

**Competing interests** SA reports Advisory Board membership for Novonordisk, Medtronic, Roche and Abbott. MJC reports personal fees from University of Sheffield. DJP reports non-financial support from Novo Nordisk, Eli Lilly and Company Limited, Abbott Diabetes Care, Sanofi-Aventis and Medtronic. SH reports Advisory board membership and consultancy for Sanofi-Aventis, Boeringher-Ingelheim, Eli-Lilly, Novo-Nordisk, Springer Medical, Zealand Pharma and UN-EEG. Personal fees for speaker panels for Novo-Nordisk. JS reports Personal fees, non-financial support and other (advisory board membership/support to attend educational meetings) from Medtronic, Roche Diabetes Care and Sanofi Diabetes, grants and personal fees from Abbott Diabetes Care, grants and personal fees from AstraZeneca.

**Patient consent for publication** Not required.

**Provenance and peer review** Not commissioned; externally peer reviewed.

**Open access** This is an open access article distributed in accordance with the Creative Commons Attribution 4.0 Unported (CC BY 4.0) license, which permits others to copy, redistribute, remix, transform and build upon this work for any purpose, provided the original work is properly cited, a link to the licence is given, and indication of whether changes were made. See: https://creativecommons.org/licenses/by/4.0/.

**ORCID iDs**
Elizabeth Coates http://orcid.org/0000-0002-2388-6102
Stephanie Amiel http://orcid.org/0000-0003-2686-5531
Michael Joseph Campbell http://orcid.org/0000-0003-3529-2739
Daniel John Pollard http://orcid.org/0000-0001-5630-0115

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
