## [Reviewer comments · BMJ Open]

ARTICLE DETAILS

TITLE (PROVISIONAL)	A protocol for a cluster randomised controlled trial of the DAFNEplus (Dose Adjustment for Normal Eating) intervention compared with 5x1 DAFNE: A lifelong approach to promote effective self-management in adults with type 1 diabetes
AUTHORS	Coates, Elizabeth; Amiel, Stephanie; Baird, Wendy; Benaissa, Mohammed; Brennan, Alan; Campbell, Michael; Chadwick, Paul; Chater, Tim; Choudhary, Pratik; Cooke, Debbie; Cooper, Cindy; Cross, Elizabeth; De Zoysa, Nicole; Eissa, Mohammad; Elliott, Jackie; Gianfrancesco, Carla; Good, Tim; Hopkins, David; Hui, Zheng; Lawton, Julia; Lorencatto, Fabiana; Michie, Susan; Pollard, Daniel; Rankin, David; Schutter, Jose; Scott, Elaine; Speight, Jane; Stanton-Fay, Stephanie; Taylor, Carolin; Thompson, Gillian; Totton, Nikki; Yardley, Lucy; Zaitcev, Aleksandr; Heller, Simon

VERSION 1 – REVIEW

REVIEWER	Amy Hess Fischl University of Chicago USA
REVIEW RETURNED	06-Aug-2020

GENERAL COMMENTS	Looking forward to seeing the outcomes! A much needed research study for T1D.
---

REVIEWER	Pamela Martyn-Nemeth University of Illinois Chicago United States
REVIEW RETURNED	13-Sep-2020

GENERAL COMMENTS	This is a very well written and complete protocol. I have only two questions/comments: 1. The trial registration date is listed as 8/5/2018; while the first enrollment is listed at 1/9/2018. Is this an error, please confirm the dates are correct. 2. Please define the abbreviation BCT (Fidelity Assessment, section 2). Thank you for the opportunity to review this paper.
---

REVIEWER	Claudio Pedone Università Campus Bio-Medico di Roma Italy
REVIEW RETURNED	14-Oct-2020

GENERAL COMMENTS	The protocol is well written and clear. I have a few issues:
--

	1) I could find no information about who will deliver the standard DAFNE intervention. Ideally, there should be a pool of people able to deliver both the DAFNE and the DAFNE plus intervention that should be randomised as well. At any rate, differences in the composition of the teams, also in terms of experience, should be highlighted as they are a potential source of bias. 2) The DAFNE plus includes a modification of the standard DAFNE protocol plus a support during follow-up and a digital technology. If I am not missing something, with the present design it will be impossible to know which of the components is actually effective and needed, and this information would be of great interest for a cost-effectiveness of the intervention. For example, the same effects of the composite intervention could be achieved only by using follow-up support, or only the DAFNE plus and digital support. 3) Please specify in the "outcome" section that the primary outcome is the difference in means of glycated hemoglobin. It would also be sensible to specify what the minimal clinically important difference is, and use this value to calculate power.
--	---

VERSION 1 – AUTHOR RESPONSE

Reviewer 1 comments

Looking forward to seeing the outcomes! A much needed study for T1D.

Thank you – it is pleasing to see that you share our enthusiasm for the DAFNEplus trial.

Reviewer 2 comments

This is a very well written and complete protocol. I have only two questions/comments:

1. The trial registration date is listed as 8/5/2018; while the first enrolment is listed at 1/9/2018. Is this an error, please confirm the dates are correct.

The dates listed for trial registration and first enrolment are presented in the UK (as opposed to the US) format. The trial was registered with the International Standard Randomised Controlled Trial registry on 8th May 2018 and the first participant was recruited to the trial on 1st September 2018. Apologies for any confusion caused.

2. Please define the abbreviation BCT (Fidelity Assessment, section 2).

Although BCT is already outlined in the intervention description section for DAFNEplus, for ease of reference, we have added the fuller meaning of this to the second paragraph of the fidelity assessment section.

Reviewer 3 comments

This protocol is well written and clear. I have a few issues:

1. I could find no information about who will deliver the standard DAFNE intervention. Ideally, there should be a pool of people able to deliver both the DAFNE and DAFNE plus intervention that should be randomised as well. At any rate, differences in composition of the teams, also in terms of experience, should be highlighted as they are a potential source of bias.

Thank you for highlighting this inconsistency between the descriptions of DAFNE and DAFNEplus. We have added the following sentence to paragraph one in the Standard DAFNE (control arm) section:

'Standard DAFNE will be delivered, as usual care, by trained DAFNE educators in the NHS, including diabetes specialist nurses, dietitians and physicians.'

As described in the 'Study setting' section of the protocol, only those Diabetes Centres with at least three DAFNE trained educators were eligible to participate in the trial. As such, we can confirm that there is a pool of people able to deliver both interventions, despite randomisation at Centre-level. In addition, we agree that potential differences in the experience and team composition between centres may provide a likely source of bias. The differences in delivery between the intervention and control centres are being evaluated as part of the fidelity assessment, to help us to understand the impact this has upon outcomes. In addition, we are collecting information on participating centre and DAFNE(plus) team characteristics and educator experience to help interpret these findings.

2. The DAFNE plus includes a modification of the standard DAFNE protocol plus a support during follow-up and a digital technology. If I am not missing something, with the present design it will be impossible to know which of the components is actually effective and needed, and this information would be of great interest for a cost-effectiveness analysis of the intervention. For example, the same effects of the composite intervention could be achieved only by using follow-up support, or only the DAFNE plus and digital support.

DAFNEplus has been designed as an integrated intervention, so we are not intending to address the cost-effectiveness of the individual components. The process evaluation (summarised in more detail in supplementary material 2) will enable us to understand which mechanisms of change impact on glycaemic control. That is, this seeks to answer the following research question: 'how do the different elements of DAFNEplus (knowledge/skills, technological, structured follow-up), individuals' interaction with these elements, and individual psychological differences trigger changes in and maintenance of key diabetes self-management behaviours?' In particular, the fidelity assessment of both DAFNEplus and DAFNE will facilitate further understanding of the extent to which the interventions are delivered as intended.

However, we do acknowledge the limitations of being unable to separate out the intervention components, but would like to highlight that the process evaluation will be important in interpreting the trial results and helping to direct us in which way we might want to further refine the intervention before implementation in practice.

3. Please specify the 'outcome' section that the primary outcome is the difference in means of glycated haemoglobin. It would also be sensible to specify what the minimal clinically important difference is, and use this value to calculate power.

The minimum clinically important difference on HbA1c which was included in the power calculation is 0.5%. The manuscript has now been amended to make this point slightly clearer, as follows:

'Using a two sample comparison of mean HbA1c at the 12-month follow-up with 2-sided alpha of 5%, a correlation of 0.5 between baseline and final values and a standard deviation of 1.45 (from previous DAFNE data), the trial sample gives 92% power to detect a 0.5% difference in HbA1c (the minimum clinically important difference) between the two treatment groups in the study.'

We have not, however, made edits to the outcome section of the manuscript, but rather, have made a point of clarification within the statistical analysis section, as follows:

'This primary analysis is to assess the difference between the two treatment groups on the mean HbA1c at 12 months which will be completed using a multiple linear regression model with coefficients estimated using generalised estimating equations (GEE) to account for the clustering design.'

We feel that this provides further clarification of the primary outcome and analysis and are grateful to the reviewer for highlighting this lack of clarity.

Editors' comments to the authors

- In the title, please state that your manuscript is a study protocol.

Thank you for highlighting this oversight. The manuscript title has been edited as follows:

'A protocol for a cluster randomised controlled trial of the DAFNEplus (Dose Adjustment for Normal Eating) intervention compared with 5x1 DAFNE: A lifelong approach to promote effective self-management in adults with type 1 diabetes'

- Please ensure that the main text contains an ethics and dissemination section as per our instructions for authors.

Apologies for this oversight. We have edited the manuscript so that there is now a single 'ethics and dissemination' section.

We hope that you find these edits to the manuscript acceptable, and will reconsider this for publication in BMJ Open.

VERSION 2 – REVIEW

REVIEWER	Claudio Pedone Università Campus Bio-Medico di Roma
REVIEW RETURNED	Claudio Pedone Università Campus Bio-Medico di Roma
GENERAL COMMENTS	The authors have satisfactorily addressed my queries, I have no further issues.